# Immediate Effects of Fine-Motor Training on Coordination and Dexterity of the Non-Dominant Hand in Healthy Adults: A Randomized Controlled Trial

**DOI:** 10.3390/bs12110446

**Published:** 2022-11-12

**Authors:** Chanhyun Park, Hohee Son

**Affiliations:** Department of Physical Therapy, College of Health Science, Catholic University of Pusan, Busan 46265, Korea

**Keywords:** fine-motor, coordination, dexterity, non-dominant hand, chopsticks

## Abstract

Several studies have demonstrated the beneficial effects of mirror training; however, only a few studies in Eastern countries have investigated fine-motor exercises using chopsticks, which have numerous advantages. We aimed to compare changes in coordination and dexterity of the non-dominant hand in healthy adults after conducting fine-motor training with the dominant hand using a mirror. We divided 100 healthy adults (age: 20–40 years) into experimental and control groups (each n = 50). The experimental group placed the non-dominant hand in a mirror box and indirectly imitated the fine-motor exercises conducted with the dominant hand using chopsticks. The control group performed the task with the non-dominant hand using chopsticks. We conducted the Chopsticks Manipulation Test and the Purdue Pegboard Test to assess the pre- and post-intervention coordination and dexterity of the non-dominant hand. Both groups showed a significant post-intervention improvement in coordination and dexterity (*p* < 0.01). There was no significant between-group difference in the functional improvement of coordination and dexterity (*p* > 0.05). Fine-motor training using mirrors and chopsticks significantly improved coordination and dexterity of the non-dominant hand. This training could be used to improve activity in brain regions associated with the non-dominant hand in healthy adults.

## 1. Introduction

In daily life, most individuals predominantly use one hand, which determines whether an individual is right- or left-handed. Asymmetric behaviour of frequently using the dominant hand more than the non-dominant hand decreases the relative dexterity and training level of muscles in the non-dominant hand as well as the cortical excitability of the non-dominant motor cortex [1]; moreover, Sperry [2] suggested that asymmetric use of the dominant hand causes uneven use of the verbal, analytical, inferential, partial, conscious, temporal, and continuous functions in the left brain hemisphere as well as the non-verbal, conceptual, intuitive, total, and unconscious functions in the right brain hemisphere. In other words, using both dominant and non-dominant hands allows even use of different brain regions.

Hand-related brain cortical excitability can be increased through mirror training. This training method involves visual feedback and was first introduced to treat phantom pain in patients with upper extremity amputation [3]. In this training method, positive visual feedback involving observation that the dominant hand movement reflected on the mirror is the movement of the non-dominant hand facilitates functional restoration of the non-dominant hand [4]. Mirror training has also been used for patients with neurological damage such as stroke [5] and athletes with musculoskeletal injuries [6]. Although the beneficial effects of this training method have been reported, the mechanisms underlying the motor-function improvement and pain relief remain unclear [7], with the mirror neuron system being considered to be neurologically related [8]. Vogt et al. [9] reported that the mirror nervous system causes stronger activation of the lower parietal cortex and ventral premotor cortex during observation of non-practised actions compared with the observation of practised actions. This suggests that the mirror neuron system is crucially involved in the early stages of imitation learning.

The mirror neuron system provides important evidence regarding the motor simulation theories of cognition and direct matching hypothesis. Specifically, it suggests that the observed motion is directly and automatically mapped in the related motor schemata. The mirror neuron system preferentially activates the premotor and parietal networks when observing or executing motions directly related to an object. This activation is not observed for motions that are not directly related to an object [10]. Specifically, for strong activation of the mirror neuron system, it is ideal to induce unfamiliar motions directly related to an object during the early learning stages. Based on this theory, Wang et al. [11] reported that movement mirroring induced additional activation in the primary and higher-order visual areas of the contralateral limb viewed by healthy adults. A functional magnetic resonance imaging (fMRI) study on healthy individuals and patients with stroke conducted by Shinoura et al. [12] found that motion mirroring of the unaffected hand acted as visual feedback for the affected hand, which resulted in extracerebrellar activation and activation of the ipsilateral primary motor area; furthermore, mirror learning yields significant hand motor recovery. Additionally, mirror training increases the participant’s interest in improving task performance [13]; moreover, it is a simple, easy and inexpensive intervention that can be quickly conducted [14,15]. Accordingly, there is active research on the effects, mechanisms, and evidence regarding mirror training [4].

Hand manipulation is essential for achieving higher levels of movement in daily life. In manipulative action, the force of each fingertip is attributable to forces by extrinsic and intrinsic muscles; further, it requires adequate coordination and control [16]. To improve hand manipulation, Sawamura et al. [17] emphasized the need for continuous chopsticks exercise, which requires complex and delicate motor skills of the hand and upper extremities. In China, chopsticks have been used since the 4th or 5th century B.C. and are mainly used for meal intake in Asian cultures [18,19]. Using chopsticks requires fine and delicate control as well as extensive use of the dominant hand. Fine-motor skills are required to use chopsticks, including active muscle control, concentration, and visual and movement coordination [20]. Specifically, fine-motor skills and eye–hand coordination required for chopsticks are essential in numerous daily activities, including tying a tie, playing the guitar, tying shoelaces, and eating using tableware [21]; furthermore, depending on the skill level, chopsticks can be used to grab, move, plug, and cut, which significantly requires fine-motor control of the hand; therefore, using chopsticks using the non-dominant hand may facilitate the enhancement of coordination and agility [19]. Additionally, it is a good strategy for increasing cortical excitability in the non-dominant motor cortex [17]. Although using chopsticks by the dominant hand and the non-dominant hand involves various hand movements, there have been only a few Asian studies and even these have often been limited to the dominant hand.

Therefore, this study aimed to assess functional changes in the non-dominant hand among healthy adults through fine-motor exercises using a mirror, not directly using tools. This study could provide the basis for fine-motor intervention involving the combination of mirrors and chopsticks.

## 2. Study Design and Methods

### 2.1. Study Design

This study was conducted as a randomized controlled experiment. Participants randomly placed in the experimental group and the control group performed a task that took about 30 min and immediately changed their coordination and dexterity through evaluation before and after performing the task. The study was conducted in the rehabilitation room of the Keonsol medical hospital in Busan, South Korea, from 1 January–20 February, 2022 and was approved by the Research Ethics Committee of the Catholic University of Pusan (CUPIRB-2021-019).

### 2.2. Participants

We recruited 108 healthy adults aged 20–40 years who had no difficulties in following the researcher’s instructions, who had sufficient muscle strength to conduct the given task, and who were without limitations in the range of motion of the joints. We excluded individuals with pain in the hand joints and muscles or other neuropsychiatric problems within 6 months before evaluation. The sample size was calculated using the G*power program, with an effect size of 0.60, significance level (α) of 0.05, and power (β) of 0.8 based on a previous study [22]. Finally, we recruited 108 participants, considering a drop-out rate of 20% with a minimum sample size of 90 [23]. Participants were randomly assigned to the experimental and control groups using 108 beads with the same weight and size in two different colours, with each participant being asked to draw a bead from a bag.

### 2.3. Intervention Method

#### 2.3.1. Experimental Group

Each participant was asked to sit on a comfortable chair with the trunk straightened and both arms placed on a table. The dominant hand held the chopsticks while the non-dominant hand was placed in a mirror box (35 cm × 24 cm × 24 cm, Figure 1) to be unseen, without chopsticks. The non-dominant hand in the mirror box imitated the movement of the dominant hand with the chopsticks (Figure 2 and Figure 3).

The participants used the dominant hand to move 10 red beans from one bowl to another bowl placed 20 cm away. In the case where the red bead was placed elsewhere than the bowl, the bead was returned to the original bowl and the task was repeated. The task of moving all the red beans was repeated five times, with a 3 min break between the tasks. The task was performed on this one occasion. The task was performed after the researcher provided a sufficient explanation for the participants to fully understand how to perform the task, and an assistant researcher who was well aware of all the contents of the task was supervising the participants’ task performances.

#### 2.3.2. Control Group

Each participant was asked to sit on a comfortable chair with the trunk straightened and both arms placed on a table. Subsequently, they used the non-dominant hand to hold the chopsticks and move 10 red beads from one bowl to another placed 20 cm away. If the red bean was placed elsewhere than the bowl, the bean was returned to the original bowl and the task was repeated (Figure 4). The task protocol was similar to that in the experimental group.

### 2.4. Research Tools

In order to confirm the immediate effect, not the long-term adaptation effect, the measurement was conducted immediately before and after the task was performed. The Chopsticks Manipulation Test (CMT) and the Purdue Pegboard Test (PPT) were conducted to evaluate hand coordination and dexterity, respectively. If a sudden event, such as unexpected pain or sneezing, that hindered the performance of the task occurred, the task was immediately stopped when the participant requested it. After sufficient rest, the task was repeated or the participant was excluded. The task was carried out after the researcher provided sufficient explanation for the participants to fully understand how to perform the task, and the assistant researcher, who was well aware of all the contents of the test, measured the participant’s test.

#### 2.4.1. Coordination Assessment: Chopsticks Manipulation Test (CMT)

The CMT was developed by Chang et al. [24] to measure the skill level of using chopsticks. Participants were asked to move 10 red beans from a container near them to another container 20 cm away. If red beans were dropped outside, not inside, the container, the process of moving them back to the original container and moving them to another container was repeated. The time it took to move all the beans was measured. Coordination can be evaluated through chopsticks control. We recorded the time it took for all the beans to be moved using chopsticks with the non-dominant hand (Figure 5).

#### 2.4.2. Dexterity Assessment: Purdue Pegboard Test (PPT)

This test was developed by Tiffin [25] to assess hand dexterity. This test allows measurement of speed and accuracy by picking up, controlling, and placing a small peg in a hole. This test comprises five items as follows: right hand, left hand, both hands, right + left hand + both hands, and assembly. We recorded the time it took for the peg to be placed in all 25 holes using the non-dominant hand.

### 2.5. Data Analysis

Statistical analyses were performed using IBM SPSS Statistics for Windows version 22.0. Statistical significance was set at α < 0.05. An independent sample t-test was conducted for between-group comparisons of differences between pre- and post-intervention values. A paired-sample t-test was conducted for intra-group comparisons of differences between pre- and post-intervention values.

## 3. Results

### 3.1. General Characteristics of Participants

Among the 108 included individuals, 100 participants (55 men and 45 women) were included in the final analysis, with eight participants being excluded. The reasons for excluding 8 out of 108 participants were as follows: two people failed to meet the meeting time even after several adjustments, three people refused to re-run it due to hand pain during the arbitration, and three people did not participate in the arbitration because they changed their minds after the group was divided. The mean age, height, and weight were 30.13 ± 4.00 years, 169.46 ± 8.75 cm, and 67.13 ± 13.57 kg, respectively; additionally, 96 and 4 participants were right-handed and left-handed, respectively. Table 1 shows the characteristics of each group. There were no significant between-group differences in the general characteristics (*p* > 0.05) (Table 1).

### 3.2. Changes in the Coordination of the Non-Dominant Hand

There were no significant between-group differences in the pre-intervention CMT time (experimental, 59.62 ± 15.02 s; control, 61.91 ± 15.14 s; *p* > 0.05) and post-intervention CMT time (experimental, 51.67 ± 9.73 s; control, 50.30 ± 14.50 s; *p* > 0.05). In the experimental group, the CMT time significantly improved from 59.62 ± 15.02 s to 51.67 ± 9.73 s (*p* < 0.05); similarly, the CMT time in the control group significantly improved from 61.91 ± 15.14 s to 50.30 ± 14.50 s (*p* < 0.05) (Table 2).

### 3.3. Changes in the Dexterity of the Non-Dominant Hand

There were no significant between-group differences in the pre-intervention PPT time (experimental, 58.13 ± 10.86 s; control, 58.30 ± 7.80 s; *p* > 0.05) and post-intervention PPT time (experimental, 52.80 ± 6.31 s; control, 54.58 ± 6.23 s; *p* > 0.05). In the experimental group, the PPT time significantly improved from 58.13 ± 10.86 s to 52.80 ± 6.31 s (*p* < 0.05). In the control group, the PPT time significantly improved from 58.30 ± 7.80 s to 54.58 ± 6.23 s (*p* < 0.05) (Table 3).

### 3.4. Between-Group Comparison of Differences in Pre- and Post-Intervention Values of Coordination and Dexterity

There was no significant between-group difference in the post-intervention change in the CMT time (experimental, −7.95 ± 15.88 s; control, −11.61 ± 15.36 s; *p* > 0.05). Further, there was no significant between-group difference in the post-intervention change in the PPT time (experimental, −5.34 ± 6.72 s; control, −3.72 ± 5.85 s; *p* > 0.05) (Table 4).

## 4. Discussion

This study examined the effects of fine-motor exercises using mirrors on non-dominant hand coordination and dexterity in healthy adults. We observed a significant post-intervention improvement in coordination and dexterity in both groups, without a significant between-group difference. Additionally, there was no significant between-group difference in the pre- and post-intervention coordination and dexterity.

### 4.1. Brain Activation and Using Chopsticks

Järveläinen et al. [26] examined brain activity by measuring cortical magnetic signals. Both direct hand contact and tool movements showed activity in the primary motor cortex, with goal-directed movement using chopsticks yielding stronger activation of the primary motor cortex; moreover, Morishita et al. [27] observed motor-evoked potentials during small muscle exercises using chopsticks and transcranial magnetic stimulation. In fine-motor control of the non-dominant hand using chopsticks, motor-evoked potentials were observed, with increased interhemispheric inhibition induced in the primary motor cortex. This indicates that motor activities using tools activates the mirror nervous system, which provides positive evidence for our observed effects of training with mirrors and chopsticks. Matsuo et al. [28] investigated haemodynamics using near-infrared spectroscopy during self-feeding using chopsticks with dominant and non-dominant hands. Using chopsticks with the dominant hand yielded significantly more vivid motor imagery than with the non-dominant hand. This suggested that the motor imagery task involving skilled control of chopsticks allowed greater vividness than that involving unskilled control as well as affecting the haemodynamic response. Specifically, this means that the task contributes to brain activity. Numerous studies have investigated motor imagery during simple tasks; however, motor imagery during complex tasks (including self-feeding with chopsticks) in clinically meaningful daily activities remains unclear. This suggests a theoretical background for our observed improved coordination and dexterity of the non-dominant hand and indicates the clinical significance of our findings. In our study, the experimental and control groups performed relatively familiar hand movements using the dominant and non-dominant hands, respectively. Kirby et al. [29] conducted an fMRI study to compare activation by the dominant and non-dominant hands in normal adults. They found that the non-dominant hand group recruited more visual and motor brain domains than the dominant hand group during a joystick task; additionally, training of the non-dominant hand yielded relatively greater brain activity than training the dominant hand. Contrastingly, training of the dominant hand yielded bilateral stimulation, long-term persistence, and after-effects. These results are suggestive of the theoretical background of our findings, where both groups showed a significant post-intervention improvement in coordination and dexterity.

### 4.2. Effects of Fine-Motor Training Using a Mirror

Matthys et al. [30] conducted an fMRI study on neural activation of the mirror neuron system during hand movements. They created an optical illusion of left-hand movement by projecting the right hand onto a mirror. They observed activation of the right superior temporal and occipital gyruses when the mirror was used, despite no left-hand movement. These brain regions are high-dimensional visual areas that analyse biological stimuli and are activated by observing biological movements. In our study, the experimental group showed significant post-intervention improvements even though the non-dominant hand did not directly perform the task. Taken together, these findings provide a theoretical basis indicating that the projection for the dominant hand through the mirror was recognized as an optical illusion of movement in the brain area of the non-dominant hand. Accordingly, this led to brain activation, which improved hand coordination and dexterity. Reflected mirror images of one upper extremity trigger the same visual stimuli as those triggered by the movement of the other upper extremity. This activates the premotor cortex by replacing the proprioceptive sensory information [5]; moreover, observing, imaging, and executing a motion involves the use of mirror neurons, which increases the excitability of the primary motor area cortex [4].

### 4.3. Effects of Fine-Motor Training with Non-Dominant Hand

Munn et al. [31] reported that cross-training of one extremity affected the other extremity or other non-exercising body parts, which promoted muscle activity. Additionally, Carson [32] reported increased motor domain excitability of the corresponding cerebral hemisphere with the use of the dominant upper extremity for training and improved function. Carroll et al. [33] showed that this mechanism is structured in a hierarchical and loose manner within an extensive network distributed through the frontal lobe of the cerebral cortex, which is involved in the planning and execution of autonomous movement. Furthermore, they showed that unilateral motor learning is connected to the brain stem through the corpus callosum between motor areas, which affects motor learning on the other side. Taken together, mirror training could lead to the effects of cross-training on the directly used dominant hand, which increases the excitability of the contralateral cerebral hemisphere. This, consequently, influenced the activation of the non-dominant hand and improved coordination and dexterity.

### 4.4. Implication

Although we observed no significant between-group difference in the functional improvement, the mirror training allowed more comfortable practice of unfamiliar tasks and increased the participants’ interest in the task. Performing the familiar task using the dominant hand reflected in the mirror facilitated imitation learning of the unfamiliar motion in the non-dominant hand. During the early stages of imitation learning, the mirror neuron system may have strongly worked to increase brain cortical activation. Furthermore, observing the mirror-reflected motion induces activation not only in higher-order visual regions but also in regions traditionally dedicated to motor function [34]. This could have attributed to easier movements by our participants. In the experimental group, the use of mirrors made it easier to train to improve the function of the non-dominant hand in an indirect way, which showed statistically significant differences. Of course, there was no statistically significant difference compared to the control group directly trained on the non-dominant hand, but it is worth noting that about 30 min of training with chopsticks performed just once significantly improved immediate functionality in both indirect and direct fine-motor training groups. This means that it is a worthwhile training method to quickly gain results in clinical practice, and it suggests that the use of chopsticks limited to Oriental culture along with the currently widely used mirror training is also appropriate training to improve function.

### 4.5. Limitations

This study has several limitations. First, we included a limited number of healthy adults aged 20–40 years; therefore, our findings cannot be generalized for patients across different ages. Second, participants differ in the degree of ability to use non-dominant hand according to their individual lifestyles. This is because the ratio of using dominant and non-dominant hands differs according to each individual’s specialized lifestyle. In addition, this study recommended using standardized chopsticks holding methods as much as possible, but it is difficult to continuously control the chopstick holding method because it is a result customized by each individual over a long period of time. Finally, it is difficult to predict long-term sustainability because only the immediate training effect was evaluated and the long-term effect was not evaluated.

## 5. Conclusions

This study showed that fine-motor training using a mirror and chopsticks improved the coordination and dexterity of the non-dominant hand in healthy adults. Accordingly, we recommend fine-motor training using mirrors and chopsticks for enhancing the activity of brain regions responsible for the control of the non-dominant hand. There is a need for follow-up studies to assess the immediate and long-term effects after continuous training; additionally, future studies are warranted to evaluate the effects of a new training combination using mirrors and chopsticks on healthy adults and hemiplegic patients with bilateral upper extremity imbalance and overuse syndrome.

## Figures and Tables

**Figure 1 behavsci-12-00446-f001:**
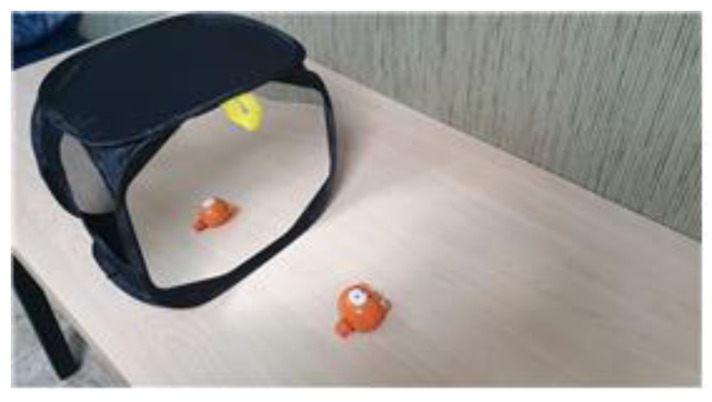
Mirror Box.

**Figure 2 behavsci-12-00446-f002:**
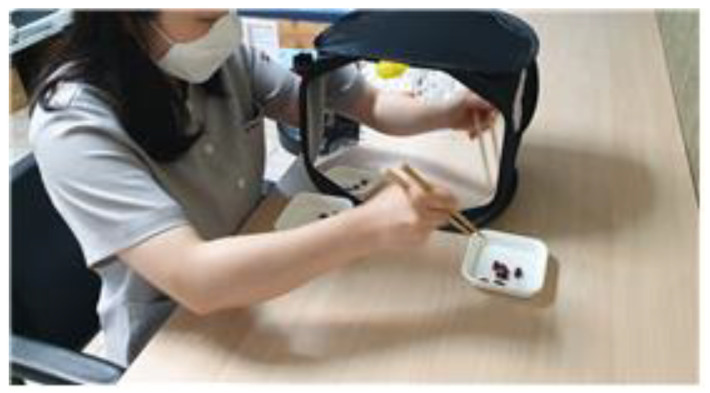
Experimental group training.

**Figure 3 behavsci-12-00446-f003:**
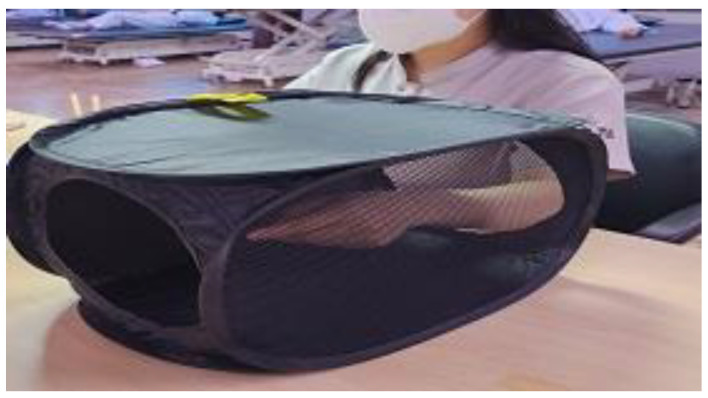
Experimental group training (the other side of the mirror box).

**Figure 4 behavsci-12-00446-f004:**
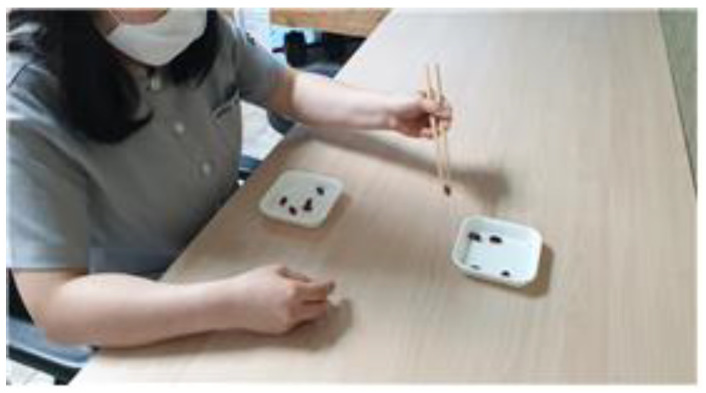
Control group training.

**Figure 5 behavsci-12-00446-f005:**
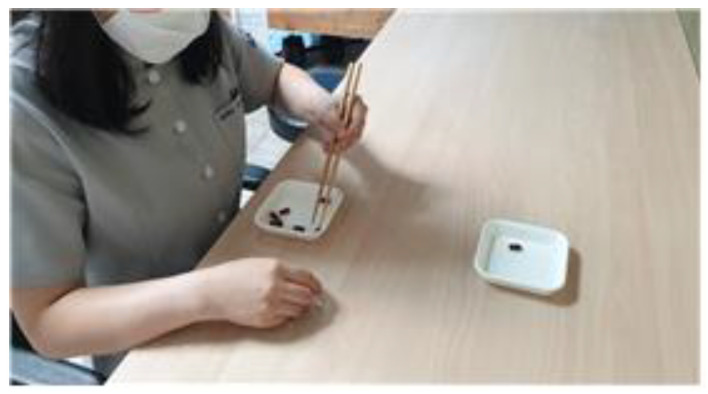
Chopsticks Manipulation Test; CMT.

**Table 1 behavsci-12-00446-t001:** Demographics. This table shows general characteristics of 100 participants in the study. Data are reported as mean ± standard deviation or count (n). EG, experimental group; CG, control group.

Characteristic	EG (n = 50)	CG (n = 50)	*p*
Gender (male/female)	28/22	27/23	0.843
Age (years)	29.94 ± 4.02	30.32 ± 4.01	0.637
Height (cm)	169.92 ± 9.70	169.00 ± 7.74	0.601
Weight (kg)	67.28 ± 12.88	66.98 ± 14.35	0.913
Dominance hand (L./R.)	1/49	3/47	0.312

**Table 2 behavsci-12-00446-t002:** Coordination. This table shows the differences in coordination between the experimental group and the control group for non-dominant hand. Data are reported as mean ± standard deviation and units are seconds. EG, experimental group; CG, control group.

	EG (n = 50)	CG (n = 50)	t	*p*
Pre	59.62 ± 15.02	61.91 ± 15.14	0.759	0.450
Post	51.67 ± 9.73	50.30 ± 14.50	0.555	0.580
t	3.541	5.346		
*p*	0.001 *	<0.001 *		

*; Significant difference (*p* < 0.05).

**Table 3 behavsci-12-00446-t003:** Dexterity. This table shows the differences in dexterity between the experimental group and the control group for non-dominant hand. Data are reported as mean ± standard deviation and units are seconds. EG, experimental group; CG, control group.

	EG (n = 50)	CG (n = 50)	t	*p*
Pre	58.13 ± 10.86	58.30 ± 7.80	0.088	0.930
Post	52.80 ± 6.31	54.58 ± 6.23	1.420	0.159
t	5.616	4.500		
*p*	<0.001 *	<0.001 *		

*; Significant difference (*p* < 0.05).

**Table 4 behavsci-12-00446-t004:** Between groups. This table shows the differences in pre- and post-performance tasks between groups. Data are reported as mean ± standard deviation and units are seconds. EG, experimental group; CG, control group, CMT; Chopsticks Manipulation Test; PPT, Purdue Pegboard Test.

	EG (n = 50)	CG (n = 50)	t	*p*
CMT	−7.95 ± 15.88	−11.61 ± 15.36	−1.172	0.244
PPT	−5.34 ± 6.72	−3.72 ± 5.85	1.282	0.203

(unit: seconds).

## Data Availability

The data and materials are available from the corresponding author upon reasonable request.

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
