# Peer review of "Immediate Effects of Fine-Motor Training on Coordination and Dexterity of the Non-Dominant Hand in Healthy Adults: A Randomized Controlled Trial"

_behavsci, 2022, doi:10.3390/bs12110446_

Round 1
Reviewer 1 Report
abstract:
13. Add "...after conducting fine-motor training with the dominant hand using a mirror"
The introduction is well-outlined and clearly explained. It draws the interest of the reader. Your topics of cortical excitability, mirror neuron system, and hand manipulation are explained well. I suggest you utilize this format for your discussion (see notes below).
The headings are mislabeled. There is no 2.0 for methods and this section is labeled incorrectly. The secondary headings italicized look less important that the tertiary headings which are not.
39: indent
76-77 combine paragraphs.
90-91: clarify this sentence- be clear that the chopsticks are used with the dominant hand only and that the non-dominant hand is copying the action without chopsticks and is out of sight. It may require a couple sentences to explain this clearly. Also recommend you add your hypothesis to this paragraph and clearly state your purpose in designing the experiment.
91-92: this sentence seems out of place and requires clarification. I think if you state your purpose and hypothesis, you will be able to connect it to the potential clinical applications. I would expand your application sentence to be more clear "new training method" does not appear to be the focus of this article.
Methods: Section 2 & 3 headings require attention in regard to formatting and title choice.
line110-115: clarify that the non-dominant hand does not have chopsticks. It would be helpful if there was a 3rd picture showing what the non-dominant hand is actually doing.
115- please state somewhere in this section that it was only done on this one occasion and that you were assessing for immediate effects only and not long-term adaptations.
139-140: this is confusing. Please give context to why the test would be stopped and why it matters. It seems drastic that a participant was excluded because they asked to stop and rest.
145-146: Please explain this test better. This description should be similar to your description of the PPT.
Table 172: Table 1 & 2 & 3 & 4-write out the headings instead of abbreviating them
Discussion: content is good with plenty of appropriate citations. Recommend re-writing discussion for clarity. Consider organizing the discussion into topics that provide by paragraph. Currently, the paragraphs lack a cohesive structure to support the main point (look at the first sentence of each paragraph and be sure the content of the paragraph clearly supports it). Recommend each paragraph have a theme sentence that connects to the purpose and the content of each paragraph is built to provide support by using previous literature to explain the reason for the results and the clinical application. I believe if you organize the discussion in this manner, it will provide focus, clarity, and clinical application of your results.
Please discuss the clinical significance of the statistical difference
276: recommend limitations is a numbered section
279-280: explain this in further detail
280: I suggest you expand on this
Please include the clinical significance of the improvement following one intervention session. How does the p-value relate to the MDC for the test? Is this a limitation? Consider addressing this in the results/discussion if not.
Author Response
Response to Reviewer 1 Comments
Point 1: 13. Add "...after conducting fine-motor training with the dominant hand using a mirror"
The introduction is well-outlined and clearly explained. It draws the interest of the reader. Your topics of cortical excitability, mirror neuron system, and hand manipulation are explained well. I suggest you utilize this format for your discussion (see notes below).
The headings are mislabeled. There is no 2.0 for methods and this section is labeled incorrectly. The secondary headings italicized look less important that the tertiary headings which are not.
Response 1: Added the contents for clear meaning delivery.
Point 2: 39: indent
Response 2: Indented.
Point 3: 76-77 combine paragraphs.
Response 3: Paragraphs are combined to fit the context.
Point 4: 90-91: clarify this sentence- be clear that the chopsticks are used with the dominant hand only and that the non-dominant hand is copying the action without chopsticks and is out of sight. It may require a couple sentences to explain this clearly. Also recommend you add your hypothesis to this paragraph and clearly state your purpose in designing the experiment.
Response 4: Added information about using chopsticks with dominant hand and non-dominant hand. The purpose of this study was explained by suggesting that there are fewer non-dominant hand papers compared to papers on Asian culture and dominant hand.
Point 5: 91-92: this sentence seems out of place and requires clarification. I think if you state your purpose and hypothesis, you will be able to connect it to the potential clinical applications. I would expand your application sentence to be more clear "new training method" does not appear to be the focus of this article.
Response 5: The training methods used in this study were written in more detail, and the sentences were modified to fit the focus of the clinical significance of this article.
Point 6: Methods: Section 2 & 3 headings require attention in regard to formatting and title choice.
Response 6: The study design was added to fit the label in the section, and the sequence was modified accordingly. The second title in italics has not been modified because it conforms to the criteria set out in this journal.
Point 7: line110-115: clarify that the non-dominant hand does not have chopsticks. It would be helpful if there was a 3rd picture showing what the non-dominant hand is actually doing.
Response 7: Added a sentence that made it clear that there were no chopsticks in the non-dominant hand, and I also added a third picture.
Point 8: 115- please state somewhere in this section that it was only done on this one occasion and that you were assessing for immediate effects only and not long-term adaptations.
Response 8: Added that the task was performed only once, and that the research tool evaluated the immediate effectiveness.
Point 9: 139-140: this is confusing. Please give context to why the test would be stopped and why it matters. It seems drastic that a participant was excluded because they asked to stop and rest.
Response 9: Changed the context that could be confusing.
Point 10: 145-146: Please explain this test better. This description should be similar to your description of the PPT.
Response 10: Contains the details of the test.
Point 11: Table 172: Table 1 & 2 & 3 & 4-write out the headings instead of abbreviating them
Response 11: The heading is not abbreviated but detailed.
Point 12: Discussion: content is good with plenty of appropriate citations. Recommend re-writing discussion for clarity. Consider organizing the discussion into topics that provide by paragraph. Currently, the paragraphs lack a cohesive structure to support the main point (look at the first sentence of each paragraph and be sure the content of the paragraph clearly supports it). Recommend each paragraph have a theme sentence that connects to the purpose and the content of each paragraph is built to provide support by using previous literature to explain the reason for the results and the clinical application. I believe if you organize the discussion in this manner, it will provide focus, clarity, and clinical application of your results.
Please discuss the clinical significance of the statistical difference
Response 12: For clarity of discussion, paragraphs were divided and topic sentences were set.
Point 13: 276: recommend limitations is a numbered section
Response 13: Added to the discussion's Implications topic.
Point 14: 279-280: explain this in further detail
Response 14: Added more details.
Point 15: 280: I suggest you expand on this
Please include the clinical significance of the improvement following one intervention session. How does the p-value relate to the MDC for the test? Is this a limitation? Consider addressing this in the results/discussion if not.
Response 15: The clinical significance of one intervention has been added to the implications of the discussion. The fact that it depends on the individual lifestyle is that there is a difference in proficiency depending on how the dominant hand and the non-dominant hand are usually used, and there are too many things to consider in order to control and conduct research. Added content for this.

Reviewer 2 Report
The study is focused on a very interesting topic, fine-motor training using a mirror and chopsticks to improve the coordination and dexterity of the non-dominant hand in healthy adults. The article is well-written and explains different concepts in detail. However, a few minor things should be corrected before it is accepted for publication.
1. The total number of participants included in this study was 100. However, the information given in the methods section is confusing, as it says 108 healthy adults were recruited in this study. It should be explained in detail how many participants participated in the study, and how many participants were excluded, and why they were excluded. The term "excluded for a personal reason" does not fit well with the empirical study unless the details of the personal reasons are provided to the readers.
2. The study limitations should be provided.
3. The implications of the study findings should be discussed, and suggestions should be provided for future studies.
4. The location and context of the study may add more value to the outcome of this study. Whether the study was conducted in China (or another country), the region inside the county.
Author Response
Response to Reviewer 2 Comments
Point 1: The total number of participants included in this study was 100. However, the information given in the methods section is confusing, as it says 108 healthy adults were recruited in this study. It should be explained in detail how many participants participated in the study, and how many participants were excluded, and why they were excluded. The term "excluded for a personal reason" does not fit well with the empirical study unless the details of the personal reasons are provided to the readers..
Response 1: Of the 108 applicants recruited, 100 were the last to participate in the study. The reasons for the excluded 8 people were explained in detail.
Point 2: The study limitations should be provided.
Response 2: Added limitations to the Discussion section.
Point 3: The implications of the study findings should be discussed, and suggestions should be provided for future studies.
Response 3: Added implications to the discussion section and also described the suggestion.
Point 4: The location and context of the study may add more value to the outcome of this study. Whether the study was conducted in China (or another country), the region inside the county.
Response 4: Added information about the country, city, and research place to the study design.

Reviewer 3 Report
The manuscript shows a fine movement training with chopsticks for the non-dominant hand in healthy subjects.
Line 60: “et al”: a full stop should be added after “al”.
Line 64: idem. Revise all citations.
Line 89: which ones?
Line 138: is there any order for these tests? Who supervise the participants? When was the recruitment process?
More information about the procedure and timing is needed.
Why did you choose healthy subjects?
What is the clinical significance of the non-dominant hand training?
Coordination exercise and therapies focus on fine training aren’t new. What is the evidence that this manuscript adds to the literature? You should highlight your results to improve the quality of the manuscript.
Author Response
Response to Reviewer 3 Comments
Point 1: Line 60: “et al”: a full stop should be added after “al”.
Response 1: All citation have been modified.
Point 2: Line 64: idem. Revise all citations.
Response 2: All citation have been modified.
Point 3: Line 89: which ones?
Response 3: Modified it so that you can clearly see that it is a measurement of non-dominant hand.
Point 4: Line 138: is there any order for these tests? Who supervise the participants? When was the recruitment process?
Response 4: It added, The researcher explained the test to the participants and the supervisor was conducted by the assistant researcher, and the assistant researcher was an employee working at the research center, so he did not go through the hiring process.
Point 5: More information about the procedure and timing is needed.
Response 5: Added a Study design and described it in more detail.
Point 6: Why did you choose healthy subjects?
Response 6: In previous literature, healthy subjects were often set up, and training using mirrors and chopsticks through healthy subjects was intended to provide evidence for clinical application to patients with upper limb discomfort due to hemiplegia or overuse syndrome.
Point 7: What is the clinical significance of the non-dominant hand training?
Response 7: The direct effect on healthy subjects is to increase coordination and dexterity, and the indirect effect is to help brain activation and induce more diverse brain regions to be used. Furthermore, it is to secure a basis for conducting research on patients suffering from upper limb discomfort.
Point 8: Coordination exercise and therapies focus on fine training aren’t new. What is the evidence that this manuscript adds to the literature? You should highlight your results to improve the quality of the manuscript.
Response 8: In this study, I wanted to show you a new training that combines chopsticks and mirrors. However, in order not to focus on training, the contents were deleted and the results were emphasized in the discussion.

Round 2
Reviewer 3 Report
Dear authors,
Thank you for considering my suggestions, discussion looks much better.
Good job.